# Graph Transformers on EHRs: Better Representation Improves Downstream Performance

**Raphael Poulain, Rahmatollah Beheshti**
University of Delaware
`{rpoulain,rbi}@udel.edu`

## Abstract

Following the success of transformer-based methods across various machine learning applications, their adoption to healthcare predictive tasks using electronic health records (EHRs) has also expanded extensively. Similarly, graph-based methods have been shown to be very effective in capturing inherent graph-type relationships in EHRs, leading to improved downstream performance. Although integrating these two families of approaches seems like a natural next step, in practice, creating such a design is challenging and has not been done. This is partly due to known EHR problems, such as high sparsity, making extracting meaningful temporal representations of medical visits challenging. In this study, we propose `GT-BEHRT`, a new approach that leverages temporal visit embeddings extracted from a graph transformer and uses a `BERT`-based model to obtain more robust patient representations, especially on longer EHR sequences. The graph-based approach allows `GT-BEHRT` to implicitly capture the intrinsic graphical relationships between medical observations (concepts), while the `BERT` model extracts the temporal relationships between visits, loosely mimicking the clinicians' decision-making process. As part of our method, we also present a two-step pre-training strategy for learning better graphical and temporal representations. We show how such improved representations can ultimately achieve state-of-the-art performance in a variety of standard medical predictive tasks, demonstrating the versatility of our approach[1].

## 1 Introduction

With the growing availability of electronic health records (EHRs), deep learning methods, capable of disentangling complex patterns in EHRs, have been widely used in various healthcare predictive tasks (Choi et al., 2016; Ma et al., 2017; Gupta et al., 2022b; 2024; Lipton et al., 2017; Choi et al., 2018; 2020). Among the numerous approaches presented in recent years in this domain, two families stand out: approaches based on the transformer architecture (Vaswani et al., 2017) and those based on graph neural networks (GNNs). Fundamentally, these two families have much in common, as both aim to learn better representations from the raw data (from a geometric deep learning view (Dwivedi & Bresson, 2021; Bronstein et al., 2021), they have the same roots). However, in practical EHR use cases, the two families have distinguishable characteristics.

*On the transformer-based side*, most methods adopt an encoder-based architecture, while the decoder-based ones are also rapidly expanding (Lee et al., 2019; Alsentzer et al., 2019; Singhal et al., 2022). Many of the encoder-based methods adopt the `BERT` (Bidirectional Encoder Representations from Transformers) model (Devlin et al., 2019) and are the ones achieving state-of-the-art performance in EHR predictive tasks, like those presented by Pang et al. (2021); Li et al. (2020); Poulain et al. (2022); Rasmy et al. (2021); Prakash et al. (2021); Meng et al. (2021); Li et al. (2021); and Chen et al. (2022). These methods generally conceptualize the medical codes (i.e., the main pieces of EHRs) as analogs to words, patient visits as sentences, and complete patient medical records as documents. This approach generates medical code embeddings while accounting for the temporal structure of EHRs through self-supervised (pre-training) strategies. However, a critical limitation arises in the encoding of the relationships between the visits, as they are only weakly represented

---

[1]Our code is publicly available at `https://github.com/healthylaife/GT-BEHRT`

via the commonly used positional encodings. Furthermore, the association between natural language words and medical codes is not straightforward. In conventional natural language, sentences inherently comprise an ordered sequence of words, while medical codes diagnosed during a single patient visit lack any temporal relationship (as they have been recorded simultaneously). Although in these methods positional tokens remain constant within each patient visit (a weak signal for interpreting all codes in one visit similarly), considering visits as words rather than sentences is a more conceptually aligned analogy. This visit-level representation views a patient's medical history as an ordered sequence of visits, akin to a sentence with visits as its constituent words.

To generate a format suitable for such a visit-level (versus a medical code-level) transformer architecture, one can generate visit embeddings using a fully connected linear layer, as what Luo et al. (2020) and Ren et al. (2021) have done. However, due to the known problems of EHRs, such as noise, missingness, and sparsity, deriving meaningful visit representation through linear layers is difficult and often fails to capture robust connections between medical codes. This is precisely where the second prominent family of EHR methods (GNN-based methods) comes in.

*On the GNN-based side*, EHRs are conceptualized as graphs, by capturing the inherent graphical relationships between various medical codes. Examples of these rich graphical relationships include the (causal or association) relationships between diagnoses and treatments, or between patients and providers. By following a graph representation learning approach, medical codes can be viewed as nodes and their relationship as edges. This graphical structure can then be used to derive meaningful patient or visit embeddings. A common concern with many previous studies of this type, however, is assuming to have prior knowledge of the structure of medical ontology trees (Shang et al., 2019; Choi et al., 2017a) or not considering the temporal patterns between the medical codes (Choi et al., 2020; Zhu & Razavian, 2021; Cai et al., 2022; Xu et al., 2023).

This study aims to reconcile these two connected but separate families of approaches. Our study runs on the hypothesis that extracting effective visit-level, graphical, and temporal representations can yield better performance in downstream predictive tasks on EHRs. Going to a coarser representation (from code-level to visit-level) also helps with the quadratic complexity of transformer-based methods Keles et al. (2022), allowing learning longer temporal patterns using our method. Our new method, `GT-BEHRT` (after, graph transformer `BERT` for EHRs), extends a graph transformer architecture to create visit embeddings used as input to a `BERT`-based encoder. To make such a hybrid design work, we propose customized steps for representation learning and pre-training. Specifically, as common pre-training approaches (like Masked Language Modeling) cannot be directly applied to our graph-based input representation, we also propose a new pre-training process. We also leverage additional contextual information for the visits (such as the patient's age and the day of the year the visit occurred) to derive more robust time-aware visit embeddings. Consequently, the contributions of our work are as follows:

- We present a novel architecture leveraging graph-based time-aware visit embeddings with a `BERT`-based model to generate patients' representations by extracting both code and visit-level information.
- We propose a tailored pre-training strategy to better capture the implicit graphical structure of EHR data and the temporal relationships between visits.
- We evaluate our method in a series of experiments and achieve state-of-the-art performance on a variety of medical predictive tasks, showing the versatility of the proposed method and its robustness to the choice of downstream predictive task.

## 2 METHOD

Our proposed method, `GT-BEHRT`, consists of two distinct steps (Figure 1). First, we extract the visit embeddings through a graph transformer-based architecture; then, we use the visit embeddings in a transformer encoder to generate patient representations used in downstream tasks.

### 2.1 INPUT REPRESENTATION

While there exists a wide range of medical information that can be stored in EHRs, in this paper without the loss of generality, we focus on three more common elements: diagnoses, medications,

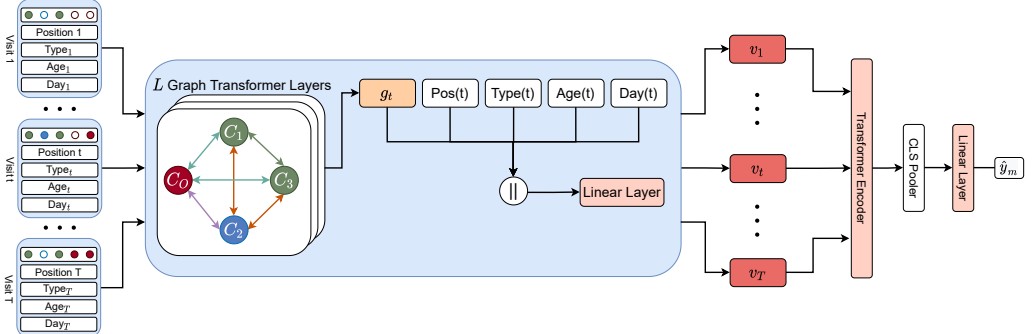

Figure 1: GT-BEHRT's architecture. The blue block describes extracting the graph visit embeddings, where $C_0, C_1, ..., C_4$ represent four exemplar medical codes associated with the visit $t$ of a fictional patient. The medical codes are color-coded to represent different code types (condition, medication, procedure). The graph is based on the medical codes associated to a given visit (visit $t$ here). Each visit goes through this process individually before being fed to the Transformer Encoder.

and procedures (our proposed method can be generalized to other cases). Therefore, we first define $\mathcal{D}, \mathcal{M}, \mathcal{P}$ as the set of all observable diagnosis, medication, and procedure codes, respectively. For simplicity, we refer to $\mathcal{O} = \mathcal{D} + \mathcal{M} + \mathcal{P}$ as the set of all observable medical codes. Let us now define the medical record of a patient $p$ as a sequence of $T$ visits, where, for each visit $t \in T$, a list of medical codes $\mathcal{C}_t^p \subseteq \mathcal{O}$ is assigned to $p$. We initially assume that all medical codes in $\mathcal{C}_t^p$ are fully connected and implicitly encode the relationship type between two codes (like, medication-medication, medication-diagnosis, etc.) as a spatial encoding. Because obtaining a single meaningful graph embedding directly from the graph node features (graph pooling) is notoriously challenging (Liu et al., 2022), we also add a virtual visit node *<VST>* and connect it to all other nodes in each visit graph with a separate edge type. This node will be used as a readout module to obtain a graph representation. This specific architecture is particularly useful in settings where the true graphical structure is unknown.

We can now note $\mathcal{G}_t^p(\mathcal{C}_t^p, \mathcal{E}_t^p)$ as the graph defining $p$'s $t^{\text{th}}$ visit, with $(e_s, e_t, e_r) = \epsilon_t \in \mathcal{E}_t$, where $e_s, e_t,$ and $e_r$ denote the source, target, and relationship type of the edge, respectively. Therefore, the graph $\mathcal{G}_t^p$ has $|\mathcal{C}_t^p| + 1$ nodes and $\binom{|\mathcal{C}_t^p|+1}{2}$ undirected edges. Additionally, we leverage other visit-level contextual information to help derive better visit embeddings. Specifically, we extract the visit type (in-patient, outpatient, ICU, ...) as well as the patient's age and the visit day of the year (1-366; to help differentiate seasonal diseases like the flu from age-related conditions such as Parkinson's or Alzheimer's diseases). Contextual information is of particular importance when analyzing patient medical records. This process allows us to better mimic the thought process used by medical professionals who would make informed decisions considering the contextual information and the uniqueness of the patient.

We can now define $\mathcal{V}_t^p = (\mathcal{G}_t^p(\mathcal{C}_t^p, \mathcal{E}_t^p), \text{Pos}_t^p, \text{Type}_t^p, \text{Age}_t^p, \text{Day}_t^p)$ as the tuple containing the graph, position of the visit in the sequence, visit type, patient's age, and day of the year for $p$'s $t^{\text{th}}$ visit. Therefore, the medical sequence for each patient $p$ can be formally defined as $\mathcal{S}_p = \mathcal{V}_1^p, \mathcal{V}_2^p, ..., \mathcal{V}_T^p$. For simplicity, we will drop the subscript $p$ and only consider one patient for the remainder of the paper. We provide a summary of the notations used throughout the paper in Table 4.

## 2.2 BERT WITH GRAPH-BASED VISIT EMBEDDINGS

To learn the hidden graphical structure of EHRs and obtain meaningful visit embeddings, we make use of our graph structure with a fully connected transformer encoder with $L$ layers. Following the message passing framework proposed by Shi et al. (2021), we modify the traditional attention mechanism used in the original transformer (Vaswani et al., 2017), and add our edge encoding $E$ to the Value and Key vectors of the neighboring nodes:

$$A_{ij} = \frac{(H_i W_Q)^\top (H_j W_K + E_{ij} W_E)}{\sqrt{d}}$$
$$\phi = \text{softmax}(A) \tag{1}$$
$$Attn_i = \sum_j \phi_{ij} (H_j W_V + E_{ij} W_E)$$

where $W_Q, W_K, W_V, W_E \in \mathbb{R}^{d \times d}$ are trainable parameters, $H$ is the node features matrix containing the feature vectors for each node in the graph, and $E_{ij}$ is a $d$-dimensional vector representing the relationship between the nodes $i$ and $j$. Note that, for simplicity, we do not show the bias terms in the linear layers throughout the paper. We repeat the process for each attention head following the original transformer paper (Vaswani et al., 2017):

$$MHA(H, E) = [\text{head}_1; \text{head}_2; ...; \text{head}_{N_h}] W_O,$$

where $W_O \in \mathbb{R}^{N_h * d \times d}$ is a learnable parameter matrix and $\text{head}_i$ is the $i^{\text{th}}$ attention head following Eq 1. We can now use the multi-head attention mechanism to update the features of the nodes for each layer $l \in L$. As proposed by Xiong et al. (2020), we perform the layer normalization ($LayerNorm$) (Ba et al., 2016) before the multi-head attention ($MHA$) and the linear layers ($LN$), which alleviates the warm-up stage issue and thus reduces the overall training time. The new node features $H_t^{l+1}$ are calculated following our updated transformer encoder architecture with residual connections:

$$\hat{H}^l = MHA(LayerNorm(H^{l-1}), E) + H^{l-1},$$
$$H^l = LN(LayerNorm(\hat{H}^l)) + \hat{H}^l, \tag{2}$$

with $H^0$ and $E$ being the nodes and edges feature vectors, obtained from the learned embeddings that map the categorical nodes and edge types to a $d$-dimensional latent space. We use the representation of the *<VST>* node of the last Graph Transformer layer (layer $L$) as a readout module to extract a graph-level embedding $g = H_{<VST>}^L$ for each visit graph. This method allows the *<VST>* node to attend to the important nodes of the graph and derive meaningful representations. We denote the entire process as $GT(\mathcal{G})$, which is a function that takes a graph as input and outputs a graph embedding.

This way, for each visit $\mathcal{V}_t \in \mathcal{S}$, we denote $g_t = GT(\mathcal{G}_t(\mathcal{C}_t, \mathcal{E}_t))$ as the graph embedding resulting from our Graph Transformer. We embed the position, visit type, age, and day of the year in $d$-dimensional vectors, following the temporal embedding process proposed in `CEHR-BERT` by Pang et al. (2021). We refer to the resulting embedding vectors as $Pos$, $Type$, $Age$, and $Day$. We can now concatenate these four vectors with $g_t$ to create a $5d$-dimensional visit representation and feed it into a fully connected layer to map it to a $d$-dimensional representation $v_t$:

$$v_t = [g_t; Pos_t; Type_t; Age_t; Day_t] W_v,$$

where $W_v \in \mathbb{R}^{5d \times d}$ is a learnable parameter matrix. We denote this entire process as Graph Visit Embeddings as shown by the blue block on Figure 1.

Following this process, we add another virtual token *<CLS>* to the visit embedding sequence and generate a sequence of visits representing each patient $p$. We feed this resulting sequence to a Transformer Encoder and denote its output sequence as $S \in \mathbb{R}^{(T+1) \times d}$ with $S = \text{TransformerEncoder}(<CLS>, v_1, v_2, ..., v_T)$. Lastly, we use the representation of the *<CLS>* token for various downstream medical tasks with a fully connected linear layer and define the loss function of the classifier $\mathcal{L}_c$ with the binary cross-entropy loss function:

$$\hat{y} = \sigma(S_{<CLS>} W_c),$$
$$\mathcal{L}_c = -y \log(\hat{y}) - (1 - y) \log(1 - \hat{y}), \tag{3}$$

where $\sigma$ is the sigmoid function and $W_c \in \mathbb{R}^{d \times 1}$ is a learnable vector.

## 2.3 PRE-TRAINING

To help our model learn better representations of EHRs, we pre-train `GT-BEHRT` in a two-step self-supervised learning process. We first pre-train the Graph Transformer only with a node-level

task, called Node Attribute Masking (NAM) (Hu et al., 2020). We then train the entire model on a combination of two tasks on a graph and temporal level: Missing Node Prediction (MNP) and Visit Type Prediction (VTP). While the first step focuses on learning node embeddings (the predictions are made using the node embeddings only, before any graph pooling operation), the second step allows us to leverage the graphical and temporal structure of EHRs, as its learning objectives are trained using the transformer's output, thus having access to the other visits in the sequence. We provide a visual description of our proposed pre-training process in Figure 4.

**Node Attribute Masking (NAM):** Inspired by Masked Language Modeling (Devlin et al., 2019) and the pre-training strategy proposed in Hu et al. (2020) for GNNs, we propose to randomly mask 15% of all nodes in each visit graph. The features representing a selected node would be replaced by those of a *[MASK]* node, and the Graph Transformer would be trained to retrieve the original nodes as shown in Figure 4 (Step 1), where, given an original visit input graph, two nodes are randomly masked (yellow nodes). We connect the node features of the last layer of our Graph Transformer $H_t^L$ to a linear layer and define NAM's loss as:

$$\mathcal{L}_{NAM} = \text{CE}(y_{NAM}, H_t^L W_{NAM}),$$

where $W_{NAM} \in \mathbb{R}^{d \times |\mathcal{O}|}$ is a learnable matrix with $\mathcal{O}$ being the set of all medical codes as defined earlier, $\text{CE}(y, x) = -\sum_{c=1}^{C} y_c \log(x_c)$ is the Cross-Entropy loss function with $C$ being the number of classes (here, $C = |\mathcal{O}|$), and $y_{NAM}$ is the original masked node. As this task is visit-dependent, the node representations $H_t^L$ do not receive any information on the previous/future visits of the sequence. This allows us to specifically learn better node representation, solely from a node-level task, before further pre-training on graph-level tasks, following the promising results in Hu et al. (2020).

**Missing Node Prediction (MNP):** Additionally, we pre-train our model on the graph and temporal levels using MNP. To do this, we randomly remove one node and its corresponding edges from every graph with three or more nodes and train our model to predict the removed node:

$$\mathcal{L}_{MNP} = \text{CE}(y_{MNP}, SW_{MNP}),$$

where $W_{MNP} \in \mathbb{R}^{d \times |\mathcal{O}|}$ is a learnable matrix and $y_{MNP}$ is the removed node. Note that, while NAM is a node-level task, MNP is a graph-level task performed using the transformer's output, after a graph pooling process.

**Visit Type Prediction (VTP):** Similar to the method proposed in `CEHR-BERT` (Pang et al., 2021), we leverage another learning objective where we randomly mask 50% of the visit types and replace them with a *[MASK]* token and train the model to retrieve the masked visit types:

$$\mathcal{L}_{VTP} = \text{CE}(y_{VTP}, SW_{VTP}),$$

where $W_{VTP} \in \mathbb{R}^{d \times |k|}$ is a learnable matrix with $k$ being the number of different visit types, and $y_{VTP}$ is the original visit type. Note that both MNP and VTP (Figure 4, Step 2) are performed using the transformer's output. Therefore, we train the model to predict the removed nodes and the masked visits given the context provided by the other visits in the sequence. The combination of the three tasks in a two-step process allows our model to be trained on both node and graph levels, as proved important in Hu et al. (2020), and on a temporal level, having access to the other visits.

## 3 EXPERIMENTS

**Data and Predictive Tasks**   We have evaluated the proposed method on a variety of predictive medical tasks on multiple datasets: in-hospital mortality (Mortality) and prolonged length of stay prediction in ICU (Length of Stay) on the MIMIC-IV dataset (Johnson et al., 2021), a dataset from the Beth Israel Deaconess Medical Center containing hospitalization data, and Heart Failure prediction on the All of Us dataset (All of Us Research Program Investigators, 2019), a large publicly-available EHR dataset of adult patients across the US. For all tasks, we extracted relevant information throughout patient visits, including conditions, medications, and procedures, as well as contextual information from the visit. Additionally, we have removed patients with less than 2 visits to leverage the temporality of patients' medical records. For brevity, we present a detailed description of cohort extractions, inclusion criteria, and dataset statistics in Appendix A.2. For preprocessing, we adopt

the standard procedure presented by Gupta et al. (2022a) for MIMIC, and by Poulain et al. (2022) for All of Us.

*Mortality:* The mortality prediction task is a binary classification task aimed at predicting the death of patients during their stay in the ICU.
*Length of Stay:* The Length of Stay task is a binary classification task in which the objective is to predict which patients will stay in the ICU for a prolonged period (more than 72 hours).
*Heart Failure:* The Heart Failure task is a binary classification task aimed at predicting the occurrence of heart failure within the next 365 days.

**Baselines**    We present several popular and state-of-the-art baseline models that use various deep-learning paradigms as reference points, providing a comprehensive evaluation landscape and a robust benchmark for our proposed method. These baselines span over several styles of methods suited for EHRs, namely, sequence-based (RNN or transformer) and GNN-based methods, allowing us to highlight the efficacy of the proposed method and the importance of combining both paradigms in downstream tasks.

`VGNN` by Zhu & Razavian (2021) is a variationally regularized encoder-decoder graph network that initially considers all medical codes to be fully connected and regularizes the node representations.
`HypEHR` by Xu et al. (2023) is a GNN-based method that considers EHRs as a hypergraph where medical codes are represented as nodes and patients as hyperedges, connecting the nodes.
`Dipole` by Ma et al. (2017) uses bi-directional RNNs with attention mechanisms. It is one of the most popular models for EHR predictive tasks, presented before the emergence of transformers.
`HiTANet` by Luo et al. (2020) is a hierarchical transformer-based model that uses local time embeddings to provide visit representations. Additionally, it reformulates the self-attention mechanism to associate timesteps with visits.
`BEHRT` by Li et al. (2020) is a `BERT`-based model that considers medical codes as words and visits as sentences. Similar to `BERT`, it leverages MLM to pre-train the model.
`CEHR-BERT` by Pang et al. (2021) extends `BEHRT` by incorporating time-aware emebddings. Additionally, it leverages the visit type in an encoder-decoder architecture and pre-trains the model using MLM (encoder) and VTP (decoder).
`CEHR-GAN-BERT` by Poulain et al. (2022) is a semi-supervised model based on `CEHR-BERT` that incorporated unlabeled data in an adversarial auxiliary task.

For all experiments, we randomly split the dataset into an 80/10/10 train/validation/test regime and repeated the process five times, each time with a different random seed. We report the average and standard deviation for each metric (AUROC, AUPRC, and F1 Score). We note that the F1 Score was computed with a threshold of 0.5, which can be suboptimal for a model not calibrated around 0.5. We provide more information on the computing environment, the implementation of the baselines and the proposed method, and some analysis of the size and training time of each model in Appendix A.3.

**Performance on the standard downstream tasks**    We report the results of our initial experiments comparing our method and the baselines on the three predictive tasks in Table 1. `GT-BEHRT` consistently outperformed all baselines tested on all metrics, with the highest improvements on the F1 Score. Interestingly, `HiTANet`, a visit-wise transformer model, narrowly outperformed `BEHRT`, a code-wise model, showing the potential of such an approach. Furthermore, `CEHR-BERT` demonstrated the best results amongst all the baselines tested, highlighting the efficacy of the time-aware embeddings and the added contextual information. It is also worth noting that the transformer-based methods outperformed other methods, including graph-based models such as `VGNN` and `HypEHR`, showing the importance of temporality in understanding medical records. Lastly, on a dataset with longer medical sequences (All of Us), visit-wise transformer models (`GT-BEHRT` and `HiTANet` showed promising results compared to code-wise transformers.

**Computational efficiency**    One major advantage of a visit-level transformer model compared to a medical code-level one is the reduced length of the input sequence. Since the number of codes observed per visit $C_t^p$ is greater than or equal to 1, the resulting sequence lengths of `GT-BEHRT` are orders of magnitude shorter compared to other `BERT`-based methods (i.e., `BEHRT` and `CEHR-BERT`), as shown in Table 5. This shorter length also translates to a larger than 70% reduction in training

| Model | Mortality | | | Length of Stay | | | Heart Failure | | |
|---|---|---|---|---|---|---|---|---|---|
| | AUROC | AUPRC | F1 Score | AUROC | AUPRC | F1 Score | AUROC | AUPRC | F1 Score |
| VGNN | 91.33 ± 0.73 | 61.87 ± 2.71 | 55.79 ± 2.26 | 80.49 ± 1.37 | 65.98 ± 1.66 | 59.64 ± 2.55 | 83.56 ± 1.52 | 56.38 ± 1.64 | 51.21 ± 1.95 |
| HypEHR | 91.08 ± 0.37 | 59.50 ± 1.03 | 54.95 ± 1.48 | 79.79 ± 1.18 | 64.94 ± 1.03 | 59.11 ± 1.48 | 82.41 ± 1.67 | 55.08 ± 1.21 | 50.93 ± 2.01 |
| Dipole | 92.24 ± 0.68 | 59.56 ± 1.20 | 56.96 ± 1.77 | 81.82 ± 1.19 | 70.55 ± 1.34 | 60.94 ± 1.31 | 86.87 ± 1.91 | 60.26 ± 2.32 | 55.34 ± 1.86 |
| HiTANet | 92.67 ± 0.32 | 62.46 ± 1.44 | 57.63 ± 2.37 | 82.04 ± 0.59 | 70.67 ± 0.30 | 60.46 ± 1.66 | 91.59 ± 1.07 | 65.29 ± 1.66 | 59.19 ± 1.41 |
| BEHRT | 92.67 ± 0.19 | 60.54 ± 1.66 | 55.76 ± 2.87 | 81.80 ± 0.52 | 69.73 ± 0.99 | 60.82 ± 1.43 | 87.07 ± 1.76 | 61.92 ± 1.64 | 54.81 ± 1.31 |
| CEHR-BERT | 93.31 ± 0.23 | 64.79 ± 2.37 | 59.22 ± 2.43 | 82.61 ± 0.31 | 71.36 ± 0.79 | 63.02 ± 0.42 | 87.37 ± 1.74 | 63.68 ± 2.82 | 57.28 ± 2.10 |
| CEHR-GAN-BERT | 93.47 ± 0.47 | 64.46 ± 1.22 | 59.07 ± 1.18 | 82.13 ± 0.61 | 71.24 ± 0.92 | 62.94 ± 1.34 | 87.65 ± 1.63 | 64.64 ± 1.67 | 57.44 ± 1.10 |
| GT-BEHRT | **94.11 ± 0.31*** | **65.55 ± 1.65** | **64.07 ± 2.13*** | **83.74 ± 0.51*** | **73.36 ± 1.19** | **65.91 ± 0.79*** | **94.37 ± 0.20*** | **73.96 ± 0.83*** | **64.70 ± 0.85*** |

Table 1: Comparison of GT-BEHRT with the baselines on two standard prediction tasks on MIMIC-IV and one task on All of Us datasets. The **boldface** indicates the best. The star symbol (*) indicates significant improvements over *all other models* using a two-tailed t-test at p≤0.01.

time (and also reduces inference time), which can, again, be explained by the quadratic nature of the attention mechanism of the transformer Keles et al. (2022).

**Performance on longer medical histories** Due to the improvement in computational efficiency, we expected that, on longer sequences, the ability to access visit embeddings rather than code embeddings would allow the models to derive better patient representations as it was the case on the All of Us dataset. To evaluate this further, we compare the performance of the baselines and GT-BEHRT by isolating patients who had 10 or more visits in the test set ($n \approx 500$).

| Model | Mortality | | | Length of Stay | | |
|---|---|---|---|---|---|---|
| | AUROC | AUPRC | F1 Score | AUROC | AUPRC | F1 Score |
| VGNN | 90.87 ± 0.39 (-0.46) | 62.41 ± 2.23 (+0.54) | 55.75 ± 2.31 (-0.04) | 80.03 ± 1.34 (-0.46) | 66.54 ± 3.61 (+0.56) | 60.44 ± 2.83 (+0.80) |
| HypEHR | 90.51 ± 0.71 (-0.57) | 59.41 ± 3.05 (-0.09) | 54.78 ± 2.84 (-0.17) | 79.17 ± 1.30 (-0.62) | 65.24 ± 2.39 (+0.30) | 59.30 ± 2.11 (+0.19) |
| Dipole | 89.95 ± 0.98 (-2.29) | 57.39 ± 4.95 (-2.17) | 54.21 ± 3.35 (-2.75) | 82.17 ± 2.59 (+0.35) | 73.07 ± 3.00 (+2.52) | 63.51 ± 3.33 (+2.57) |
| HiTANet | 89.61 ± 1.57 (-3.06) | 62.26 ± 3.91 (-0.20) | 54.16 ± 3.00 (-3.47) | 80.75 ± 1.51 (-1.29) | 68.08 ± 1.32 (-2.59) | 60.56 ± 2.99 (+0.09) |
| BEHRT | 86.32 ± 1.32 (-5.75) | 53.87 ± 4.57 (-6.67) | 49.29 ± 2.49 (-6.5) | 79.15 ± 1.67 (-2.65) | 60.61 ± 2.36 (-9.12) | 56.14 ± 1.88 (-4.68) |
| CEHR-BERT | 88.88 ± 0.84 (-4.67) | 57.31 ± 4.55 (-7.48) | 53.22 ± 3.95 (-6.00) | 81.23 ± 1.92 (-1.38) | 69.14 ± 5.06 (-2.22) | 60.65 ± 5.39 (-2.37) |
| CEHR-GAN-BERT | 88.21 ± 0.58 (-5.26) | 56.35 ± 4.61 (-8.11) | 53.79 ± 3.53 (-5.28) | 80.02 ± 0.94 (-2.11) | 69.92 ± 2.66 (-1.32) | 60.20 ± 3.09 (-2.74) |
| GT-BEHRT | **91.64 ± 0.74 (-2.47)** | **64.27 ± 3.45 (-1.28)** | **62.70 ± 2.91 (-1.37)** | **85.16 ± 1.20 (+1.42)** | **76.16 ± 2.44 (+2.80)** | **68.37 ± 2.17 (+2.46)** |

Table 2: Results on patients with 10+ visits. Mean ± Std (difference from Table 1 results).

Table 2 shows that the temporal models leveraging visits as inputs outperform code-wise methods in terms of performance drop or gain from the results on the entire testing set (the numbers in parentheses in the table). Notably, for the Mortality task, CEHR-BERT, a code-wise model that outperforms HiTANet on the entire dataset, suffers from a much greater performance decrease. A similar pattern can be seen on both Dipole and the proposed method, which shows, on average, the lowest performance degradation. While code-wise methods also mostly decreased on the Length of Stay task, Dipole and GT-BEHRT on the other hand show increased performance, highlighting the importance of capturing temporality in complex downstream tasks. We note that, ideally, the number inside the parentheses should be close to 0, meaning that the model's performance is agnostic to the length of the patient's medical history. Expectedly, VGNN and HypEHR show an unchanged performance, regardless of the length of the medical history, due to the fact that the models do not consider temporality. Overall, these results show that GT-BEHRT can serve as a well-rounded method, capable of achieving state-of-the-art performance on long medical sequences while still outperforming all baselines tested on the entire dataset.

**Subgroup analysis** To investigate the performance of GT-BEHRT on different subgroups (group fairness), we report the results of the experiments on the MIMIC-IV dataset for various genders and races/ethnicities. Table 6 shows that most results discrepancies remain within a single standard deviation. We believe fairness mitigation techniques (Wan et al., 2021) could improve these results.

**Ablation study** To study the effectiveness of each component of GT-BEHRT, we conducted an ablation study by removing the key components of our method and comparing the results of each variant. In particular, we evaluated the efficacy of our pre-training strategy and the use of the Graph Transformers to produce the visit embeddings. For the pre-training strategy, we alternatively remove one of the two stages (NAM and VTP+MNP) and compare the results, while we replace the Graph

Transformer (GT) with a simple linear layer where the input becomes a simple multi-hot encoding. We present the results of our ablation study in Table 3. Note that, because NAM is designed to work on graphical data and cannot be applied to visit embeddings obtained through linear layers, we do not report the variants with the NAM pre-training objective when removing the Graph Transformer component.

| NAM | VTP/MNP | GT | Mortality | | | Length of Stay | | | Heart Failure | | |
|---|---|---|---|---|---|---|---|---|---|---|---|
| | | | AUROC | AUPRC | F1 Score | AUROC | AUPRC | F1 Score | AUROC | AUPRC | F1 Score |
| × | × | × | $93.61 \pm 0.49$ | $64.60 \pm 2.10$ | $60.58 \pm 1.50$ | $82.14 \pm 0.48$ | $71.50 \pm 0.86$ | $62.95 \pm 0.67$ | $92.07 \pm 1.41$ | $68.59 \pm 1.29$ | $61.02 \pm 1.17$ |
| × | ✓ | × | $93.63 \pm 0.55$ | $65.30 \pm 1.81$ | $61.54 \pm 1.92$ | $82.70 \pm 0.66$ | $71.39 \pm 1.00$ | $63.92 \pm 1.45$ | $92.83 \pm 0.55$ | $70.80 \pm 1.92$ | $61.82 \pm 0.69$ |
| × | × | ✓ | $92.37 \pm 0.56$ | $61.58 \pm 1.52$ | $60.39 \pm 1.62$ | $81.79 \pm 0.90$ | $70.98 \pm 1.67$ | $61.26 \pm 1.95$ | $92.46 \pm 1.39$ | $69.03 \pm 1.55$ | $61.28 \pm 1.05$ |
| ✓ | × | ✓ | $93.65 \pm 0.69$ | $63.72 \pm 1.76$ | $62.21 \pm 1.74$ | $83.09 \pm 0.55$ | $73.01 \pm 0.77$ | $63.59 \pm 1.29$ | $93.30 \pm 0.87$ | $71.37 \pm 1.24$ | $62.46 \pm 1.79$ |
| × | ✓ | ✓ | $93.55 \pm 0.43$ | $64.29 \pm 1.52$ | $61.62 \pm 1.76$ | $83.72 \pm 0.53$ | $73.24 \pm 1.20$ | $65.10 \pm 0.81$ | $93.54 \pm 0.29$ | $71.47 \pm 1.22$ | $61.88 \pm 0.95$ |
| ✓ | ✓ | ✓ | $\mathbf{94.11 \pm 0.31}$ | $\mathbf{65.55 \pm 1.65}$ | $\mathbf{64.07 \pm 2.13}$ | $\mathbf{83.74 \pm 0.51}$ | $\mathbf{73.36 \pm 1.19}$ | $\mathbf{65.91 \pm 0.79}$ | $\mathbf{94.37 \pm 0.20}$ | $\mathbf{73.96 \pm 0.83}$ | $\mathbf{64.70 \pm 0.85}$ |

Table 3: Ablation study. ✓: Component is present, ×: Component has been removed. The **boldface** and highlighted box indicate the best and the second-best performing variants, respectively.

Several takeaways can be drawn from the ablation study. First, both pre-training strategies seem to have a great impact on the overall results, as showcased by the second, fourth, and fifth rows. However, the impact of pre-training on the model with linear visit embeddings, while showing some improvements, remains mostly within a standard deviation. Second, NAM obtained better results than VTP+MNP on the Morality task which is rather unexpected, as only the Graph Transformer is trained during NAM as opposed to the entire network. However, VTP+MNP outperformed NAM on the Length of Stay task. Lastly, the use of the Graph Transformer without pre-training can impede the model's performance. However, it benefits greatly from the knowledge acquired during pre-training, more so on average than its linear layer counterpart. It is also worth noting that both non-pre-trained models (first and third row) remain competitive compared to the baselines tested in Table 1, even the ones relying on a pre-training process.

**Graph-level interpretation** Amongst the many benefits of using graph-based visit embeddings is the possibility to interpret and visualize the model's output as well as analyze its mapping of the node to a latent space. To understand how the model interprets different medical codes, we plot a 2D visualization of the latent space of 1,000 most frequent medical codes using t-SNE (van der Maaten & Hinton, 2008) in Figure 5 a. Although the model was never explicitly told the nature of each medical code, it visibly grouped codes of the same type (diagnoses, medications, or procedures) close to each other in the latent space, signaling the model's loose understanding of the medical codes and their purpose. Additionally, Figure 5 b shows a heatmap of the attention weights from the graph transformer's last layer for one random visit. One can observe that the attention weights are not sparse, demonstrating that the model has indeed learned meaningful representations for code relationships. The sparsity of learned representations has been shown to be problematic in complex predictive medical tasks using GNNs (Zhu & Razavian, 2021; Choi et al., 2020). Lastly, 5 c shows the graph inferred by the model from the attention heatmap, where we removed the edges with an attention score $\leq 0.1$. Using graphical structures of EHRs and being able to show the connections between the medical codes can be very useful for the end-users (such as providers and practitioners) in interpreting the results of the models and increasing trust in the systems.

## 4 RELATED WORK

Here we discuss a non-exhaustive list of studies that are most relevant to the proposed method.

**Transformer-based models for EHRs** Following the advances made by the Transformer architecture (Vaswani et al., 2017) in natural language processing (NLP), researchers quickly started adopting it to longitudinal healthcare data. Most studies of this type fall into two subcategories by either considering the visits or the medical codes as equivalents to NLP's tokens. Intuitively, the former better leverages the temporal relationships of visits while the latter is better suited to capture the relationships between the medical codes. SAnD (Song et al., 2017) was an earlier study that used a 1D convolutional layer to obtain visit embeddings and kept the original transformer architecture. Similarly, HiTANet (Luo et al., 2020) and RAPT (Ren et al., 2021) both leverage the

transformer architecture and consider the visits as the tokens by creating visit embeddings with a fully-connected linear layer. `RAPT` proposed three pre-training tasks to learn robust patient representations, while `HiTANet` proposed a time-aware transformer to model time information in local and global stages without pre-training. One of the first studies that considered medical codes as tokens was the `BEHRT` (BERT + EHR) model (Li et al., 2020). Building on the success of `BEHRT`, many studies extended this architecture. To name a few, `Med-BERT` (Rasmy et al., 2021) pre-trained the model on a prolonged length of stay prediction task, while `RareBERT` (Prakash et al., 2021) was trained to identify patients with rare diseases in highly imbalanced datasets. `CEHR-BERT` (Pang et al., 2021) introduced a new embedding process using a fully-connected linear layer to create temporal embeddings, instead of simply summing all embeddings as in previous BERT-based EHR applications and `CEHR-GAN-BERT` (Poulain et al., 2022) extended the method for few-shot learning with a generative adversarial network (Goodfellow et al., 2014). `BRLTM` (Meng et al., 2021) added gender embeddings, and `BEHRT-CVD` Poulain et al. (2021) included race and ethnicity. Recently, encoder-based models have also demonstrated very promising performances on both clinical notes Lee et al. (2019); Alsentzer et al. (2019) and structured EHRs Wornow et al. (2023). To the best of our knowledge, only `DL-BERT` (Chen et al., 2022) has proposed an architecture to capture both the visit-level and code-level relationships by stacking two `BERT`-based models. However, the method implies a temporal ordering between the medical codes of a given visit.

**Graph-based methods for EHRs**    Given the recent advances in graph-based machine learning, many researchers have started investigating the graphical structure of EHRs aim to extract robust relationships within EHRs to allow for better learning of medical representations Li et al. (2022); Fouladvand et al. (2023). A common approach for such studies is to leverage prior domain knowledge through ontology trees. Notably, `GRAM` (Choi et al., 2017a) proposed to learn embeddings for a medical node through a convex combination of the embeddings of the code and its ancestors in an ontology graph. Similarly, `G-BERT` (Shang et al., 2019) used GNNs by constructing graphs from ontology trees to create code embeddings and fed them to a `BERT`-based model for each visit individually, preventing the model from truly learning the temporal information between visits. Also, Sun et al. (2021) combined the medical concept graph with the patient graph for disease prediction, and `MiME` (Choi et al., 2018) studied the hierarchical structure between diagnoses and prescribed medications. While these methods yielded promising results, they assume prior knowledge of the structure of the ontology tree which can be unrealistic in some real-world scenarios. Consequently, Choi et al. (2020) proposed to use Graph Convolution Transformer by representing a visit as a fully-connected graph between all the codes associated with the given visit, while later Zhu & Razavian (2021) used Variationally Regularized GNN (Kipf & Welling, 2016) with a similar representation on a patient-level instead of a single visit. Additionally, GNN-based methods can also be used with additional EHR data. For example, `ME2Vec` (Wu et al., 2021) expanded `node2vec` (Grover & Leskovec, 2016) to extract embeddings for patients, medical services, and doctors and Park et al. (2022) proposed a multimodal approach with a GNN to extract the graphical structure of EHR and a language model to handle clinical notes. Though these methods have been effective at capturing the underlying structure of EHRs, none of them studied their temporal dependencies. More closely related to our work, Liu et al. (2020) and Lee et al. (2020) combined GNNs with RNNs to represent static and time-dependent nodes, respectively. While these studies extract temporal information alongside graph-related information, they do not study the structure of a visit to enhance the quality of the visit embeddings, to be used later in a sequence model (such as a transformer or RNN).

## 5    CONCLUSION

In this work, we presented `GT-BEHRT`, a new `BERT`-based model that leverages Graph Transformers that can: 1) capture better graphical structure of EHRs, 2) derive better visit-level representations, 3) learn temporal patterns. Through extensive experiments on a series of standard tasks, we showed that our model outperforms other state-of-the-art baselines. Specifically, due to the ability to learn longer temporal patterns, it outperforms other baselines by a wide margin when dealing with longer medical sequences. As a general method agnostic to the choice of downstream tasks, one area for future work would be evaluating our method across more diverse tasks and datasets.

ACKNOWLEDGMENTS

Our study was supported by NIH awards, P20GM103446 and P20GM113125.

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

## A  DETAILS FOR NOTATIONS, DATASET AND IMPLEMENTATIONS

### A.1  NOTATIONS

For clarity, we include in Table 4 a reference table to the notations used throughout the paper.

Table 4: Reference table to the notations used in the paper.

| Notation | Description |
|---|---|
| $\mathcal{D}$ | Set of unique diagnoses codes |
| $\mathcal{M}$ | Set of unique medication codes |
| $\mathcal{P}$ | Set of unique procedure codes |
| $\mathcal{O}$ | Set of unique medical codes $\mathcal{O} = \mathcal{D} + \mathcal{M} + \mathcal{P}$ |
| $\mathcal{V}_t^p$ | Patient $p$ $t^{\text{th}}$ visit |
| $\mathcal{C}_t^p$ | Set of observed codes at patient $p$'s $t^{\text{th}}$ visit |
| $\mathcal{G}_t^p(\mathcal{C}_t^p, \mathcal{E}_t^p)$ | Graph representations of $\mathcal{C}_t^p$ |
| $\mathcal{S}_p$ | Patient $p$ medical sequence |
| $d$ | Embedding size |
| $L$ | Number of Graph Transformer layers |

### A.2  DATASET

The MIMIC-IV dataset Johnson et al. (2021) has been extensively used within the machine learning community. Among many possible downstream tasks Gupta et al. (2022a), the mortality in the ICU and ICU length of stay tasks are often used as benchmarks in the field. For that reason, we have extracted a cohort of patients who have been admitted to the ICU using the MIMIC-IV dataset and evaluated our models in the two aforementioned tasks. Specifically, we define the mortality task as a binary classification task where our objective is to predict the patients who will die during their stay in the ICU, that is, between admission time and discharge time. Similarly, the length of stay task is designed as a binary classification task in which the goal is to predict patients who will have a "prolonged" ($> 3$ days) length of stay in the ICU. Consequently, we have extracted all medical records of patients from the hosp module (non-ICU data) prior to admission to the ICU. That is, we extracted the conditions, medications, and procedures codes, together with additional contextual visit information (patient's age, visit type, and day of the year) that appeared before the patient's admission to the ICU. To leverage the temporality of patient data, we only include patients with two or more visits. Finally, this process results in a cohort of 22,376 patients. Additionally, we have also extracted a pre-training cohort of 76,253 patients. We provide further information on the cohort statistics in Figure 2.

Similar to Choi et al. (2017b); Poulain et al. (2022), we evaluate our method on a heart failure prediction task in patients between 18 and 85 years of age with the All of Us dataset. Specifically, our objective is to predict the occurrence of heart failure within the next 365 days using the past 3 years of medical data as input. To prevent information leakage, we predict the first occurrence of heart failure for patients with multiple heart failure events. Furthermore, for each positive patient, we randomly include up to 10 control patients with similar ages and used the same time windows for all patients. This process results in a cohort of 29,736 patients and a pre-training cohort of 95,358 patients. We provide further information on the cohort statistics in Figure 3.

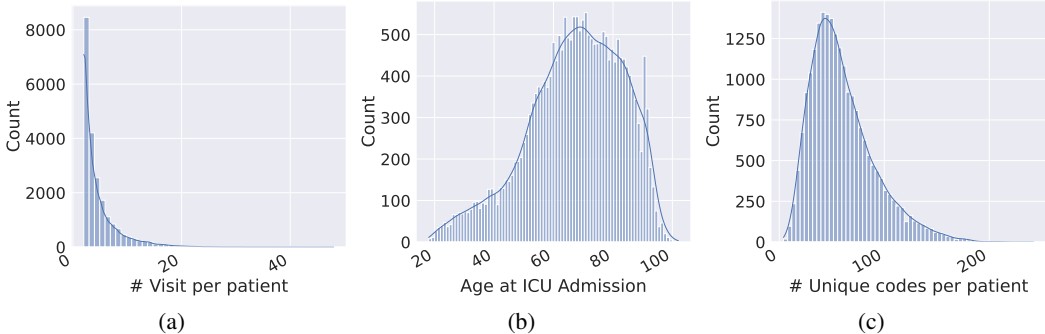

Figure 2: Key statistics of the MIMIC-IV dataset. (a): Distribution of the number of visits per patient. (b): Distribution of the patient's age at admission to the ICU. (c) Distribution of the number of unique medical codes per patient.

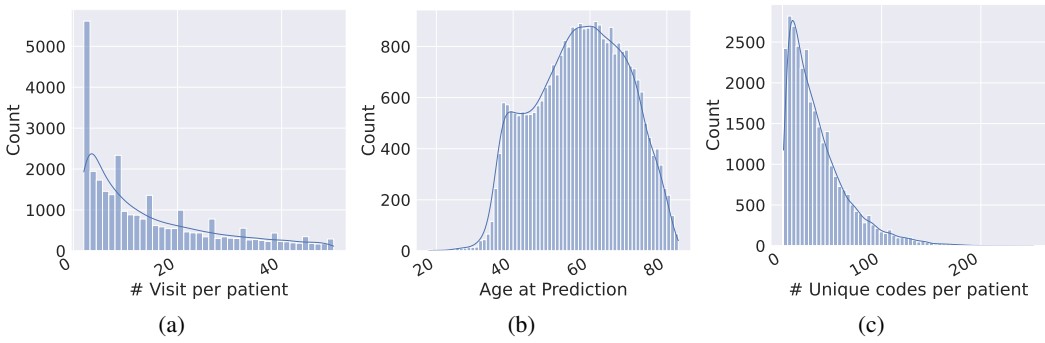

Figure 3: Key statistics of the All of Us dataset. (a): Distribution of the number of visits per patient. (b): Distribution of the patient's age at time of prediction. (c) Distribution of the number of unique medical codes per patient.

## A.3 IMPLEMENTATIONS

We have used PyTorch Paszke et al. (2019) to implement all the baselines using their respective public codebases. Additionally, we have used PyHealth's Yang et al. (2023) implementation of the Dipole Ma et al. (2017) model. We implemented the graph component of `GT-BEHRT` using PyTorch Geometric Fey & Lenssen (2019) and all the experiments were run on an NVIDIA T4 GPU. For each baseline, as well as the proposed method, we have determined the optimal hyperparameters through grid search. We report a complexity analysis of the models as well as the hyperparameters used throughout the experiments in Table 5. Note that, while VGNN (Zhu & Razavian, 2021) is a graph-based method, the official code is not written using a PyTorch graph library, therefore having significantly slower runtimes.

| | VGNN | HypEHR | Dipole | HiTANet | BEHRT | CEHR-BERT | CEHR-GAN-BERT | GT-BEHRT |
|---|---|---|---|---|---|---|---|---|
| Total number of trainable parameters | 7.3M | 270k | 9.5M | 11.7M | 11.6M | 12.7M | 13.9M | 13.7M |
| Training Time per 1,000 patients (s) | $40.31 \pm 1.55$ | $1.83 \pm 0.28$ | $2.57 \pm 0.07$ | $4.09 \pm 0.95$ | $25.23 \pm 0.92$ | $27.21 \pm 0.36$ | $41.48 \pm 0.61$ | $7.63 \pm 1.02$ |
| Inference Time per 1,000 patients (s) | $13.73 \pm 1.11$ | $1.36 \pm 0.27$ | $0.97 \pm 0.03$ | $1.89 \pm 0.13$ | $8.21 \pm 0.44$ | $8.77 \pm 0.67$ | $9.73 \pm 0.41$ | $4.39 \pm 0.51$ |
| Sequence Length | - | - | $4.90 \pm 5.08$ | $4.90 \pm 5.08$ | $102.98 \pm 81.77$ | $108.39 \pm 84.73$ | $108.39 \pm 84.73$ | $4.90 \pm 5.08$ |
| Number of Graph Layers | 2 | 2 | - | - | - | - | - | 3 |
| Number of Graph Attention Head | 3 | 4 | - | - | - | - | - | 2 |
| Graph Hidden Size | 256 | 64 | - | - | - | - | - | 108 |
| Number of Transformer Layers | - | - | 2 (RNN) | 6 | 6 | 6 | 6 | 6 |
| Number of Transformer Attention Heads | - | - | 4 (RNN) | 8 | 12 | 12 | 12 | 12 |
| Transformer Hidden Size | - | - | 512 (RNN) | 400 | 540 | 540 | 540 | 540 |

Table 5: Running time statistics for each baseline and `GT-BEHRT` with their respective hyperparameters. "-" means that it is non-applicable to the given method

# B   PRE-TRAINING

To ensure the model acquires meaningful and transferable knowledge through pre-training, we first pre-train our graph on a node-level, that is, the task are based on the nodes themselves as opposed to the entire graph. Then, we complete the pre-training process on a graph and temporal level. The tasks, realized after the transformer network, now involve the graph embeddings (as opposed to the nodes) and have full access of the context surrounding them (future/past visits). This enables us to teach the model to derive meaningful visits embeddings through the graph, and the temporality of EHRs with the transformer encoder.

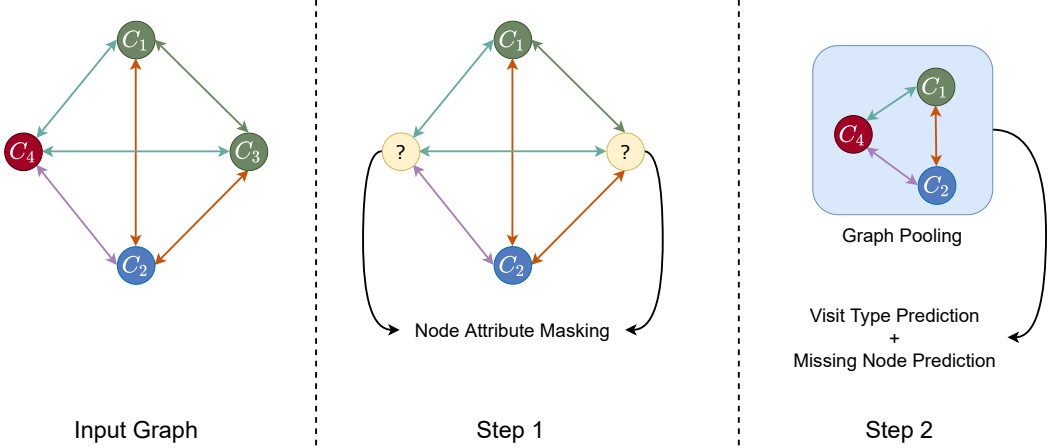

Figure 4: Our two steps pre-training process with a given input graph (left). The code notation and color coding is the same as the ones used in Figure 1. The first step randomly masks nodes (yellow nodes) to be retrieved by the model. The second step randomly removed a single node and the model is trained to retrieve the missing node while predicting the type of visit. Step 1 solely pre-trains the graph transformer through Node Attribute Masking. The second step pre-trains the entire model with Missing Node Prediction and Visit Type Prediction.

# C   ADDITIONNAL EXPERIMENTS

| Model | Mortality | | | Length of Stay | | |
|---|---|---|---|---|---|---|
| | AUROC | AUPRC | F1 Score | AUROC | AUPRC | F1 Score |
| Male | $94.35 \pm 0.63$ | $66.34 \pm 2.99$ | $65.55 \pm 2.89$ | $84.34 \pm 1.34$ | $75.43 \pm 2.44$ | $67.09 \pm 1.32$ |
| Female | $93.75 \pm 0.82$ | $64.59 \pm 2.74$ | $63.09 \pm 2.55$ | $82.86 \pm 0.35$ | $71.71 \pm 0.86$ | $63.69 \pm 1.11$ |
| White | $93.85 \pm 0.32$ | $64.91 \pm 2.69$ | $63.74 \pm 1.69$ | $83.55 \pm 0.56$ | $74.09 \pm 0.70$ | $65.39 \pm 1.44$ |
| Black/African American | $93.01 \pm 0.75$ | $65.37 \pm 2.90$ | $63.62 \pm 3.43$ | $84.96 \pm 2.42$ | $75.54 \pm 2.95$ | $67.43 \pm 2.73$ |
| Hispanic/Latino | $95.89 \pm 0.87$ | $68.45 \pm 3.64$ | $68.41 \pm 2.36$ | $84.08 \pm 2.83$ | $73.18 \pm 2.00$ | $63.74 \pm 2.77$ |
| Asian | $92.69 \pm 0.85$ | $69.25 \pm 2.44$ | $68.67 \pm 4.03$ | $81.58 \pm 1.73$ | $70.81 \pm 3.28$ | $60.84 \pm 3.03$ |
| Other | $95.19 \pm 0.53$ | $65.24 \pm 3.32$ | $64.32 \pm 3.62$ | $83.73 \pm 2.49$ | $73.12 \pm 2.34$ | $66.25 \pm 1.50$ |
| Aggregate | $94.11 \pm 0.31$ | $65.55 \pm 1.65$ | $64.07 \pm 2.13$ | $83.74 \pm 0.51$ | $73.36 \pm 1.19$ | $65.91 \pm 0.79$ |

Table 6: Comparison of `GT-BEHRT`'s results on the MIMIC-IV dataset across different demographic subgroups.

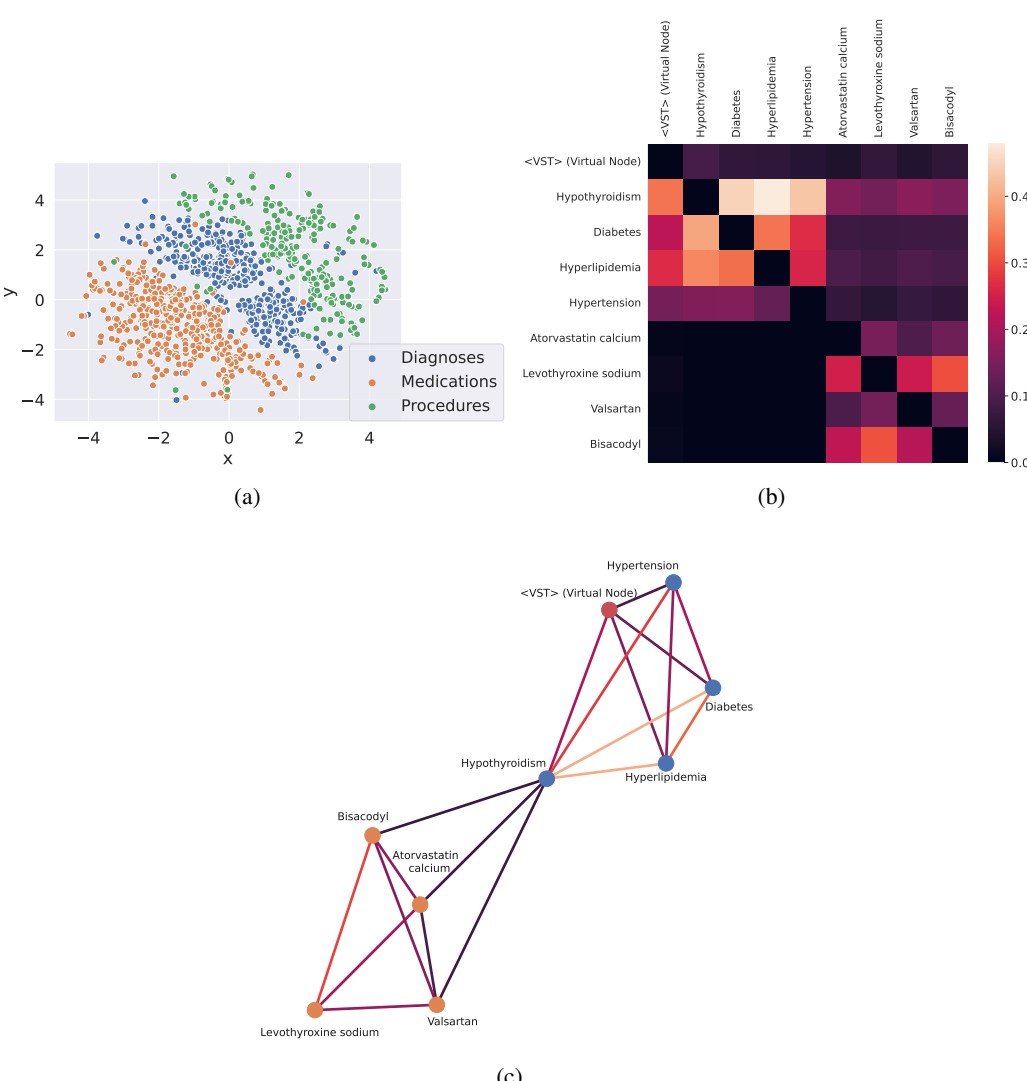

Figure 5: (a): 2D visualization of the medical codes embeddings space using t-SNE. (b): Heatmap of the attention weights from the graph transformer's last layer for a randomly selected visit. (c): Inferred graph from the attention weights displayed in (b).

