# OpenReview forum: "Graph Transformers on EHRs: Better Representation Improves Downstream Performance"
_ICLR.cc/2024/Conference — ICLR 2024 poster_

### Official Review · Reviewer_CybP · 2023-10-29

**Soundness:** 3 good
**Presentation:** 3 good
**Contribution:** 2 fair
**Rating:** 5
**Confidence:** 4

**Summary:**

The paper proposes a BERT-based model to represent EHR with a graph-based time-aware visit embedding to better capture the implicit graphical structure of EHR data and the temporal relationships between visits. The paper shows better performances in multiple tasks on the MIMIC-IV dataset.

**Strengths:**

- The paper demonstrates the importance of temporal effects in the EHR. This is insightful for research on EHR.

- Comprehensive studies are conducted including various tasks and ablation settings. The confidence intervals are provided.

**Weaknesses:**

- The proposed method is a combination of building blocks originated from several previous papers, e.g. graph transformer, BEHRT. It is a good application paper that utilizes all these aspects, but the methodological impact is limited.

- In the ablation study section, the author claims that "both pre-training strategies seem to have a great impact on the overall results, as showcased by the second, fourth, and fifth rows.". However, in fact, Table 3 shows the performance differences between experiments with and without NAM, MNP, and GT are marginal. The confidence interval of the last row and the first row (the simplest version) even overlap. The simple linear model seems to already perform very well compared to GT. A better way of justifying the impact of pretraining can be directly using linear probing on the frozen pre-trained representation.

- The power of pretraining seems to be limited by the size of the dataset (around 20K patients). Under this condition, pretraining a large model may not significantly outperform a simple linear model. It will be more exciting to apply the proposed method to a larger dataset.

**Questions:**

- How is the F1 score calculated? Which threshold is picked? Some experiments with similar AUPRC result in substantially different F1 scores.

- What is the difference between NAM and MNP? It seems they can be merged into one loss.

---

> ### Author Response · Authors · 2023-11-23
>
> **Good application paper, limited methodological impact**
> While our study, at its core, is application-driven (i.e., EHR analysis), we believe it presents several fundamental algorithmic contributions. A key contribution is presenting an effective method to extract visit-level embeddings for the transformer-based methods, instead of the common code-level embeddings. Making such pipeline work required many engineering and design considerations, especially using a graph-based approach. We also show that our pipeline helps with the quadratic complexity of transformer-based methods. Moreover, we present two new pretraining methods, that are specifically tailored to our design. We have made adjustments to the Intro to more clearly present our technical contributions.
>
> **Ablation study clarification**
> We agree that the linear model already performs well in comparison to the other baselines, however, we believe that the impact of pre-training should be compared using the same architecture baseline. That is, one should compare the results of the linear model with pre-training (2nd row) and without pre-training (1st row), and compare the GT model with pre-training (4th/5th row) and without (3rd row) pre-training. By doing so, it can be seen that when using GT, 8 of the 12 reported metrics using the pre-training outperform the GT baseline (these differences are also statistically significant using a t-test). Additionally, we believe that the true impact of the pre-training process is better highlighted by the final version of the model, which uses the entire two-step pre-training process. Nevertheless, we agree that the impact of pre-training on the linear model, while showing some improvements, remains marginal and we have made this aspect clearer in the manuscript.
>
>
> **Use a larger dataset**
> We have now run all experiments on a large cohort of the All of Us dataset (a very large dataset of ~1 million individuals). The Experiments section is updated accordingly.
>
>
>
> **F1 score clarification**
> For all experiments run, the F1 scores were calculated using the “torcheval” library, using the standard formula and a threshold of 0.5. The disparity in the F1 scores can probably be explained by a poorer calibration of the models where the threshold of 0.5 is suboptimal. We added a short note to the Experiments section about this.
>
> **NAM vs MNP**
> While NAM and MNP share some similarities, they are two different tasks. NAM is in essence similar to Masked Language Modeling, where for each graph, we randomly mask one or more nodes and use the nodes’ neighborhoods to retrieve the original nodes’ attributes. In MNP, we remove one node from the graph and use the graphs’ embeddings to retrieve the missing nodes. Thus NAM is a node-level task, while MNP is a graph-level task. Additionally, NAM is performed using the Graph Transformer only, while MNP uses the entire network, having access to the past/future visits. To address the confusion, we have modified Figure 4 and changed the description of this section to make more references to the illustration for better understanding.

---

### Official Review · Reviewer_efXj · 2023-10-31

**Soundness:** 3 good
**Presentation:** 2 fair
**Contribution:** 3 good
**Rating:** 8
**Confidence:** 5

**Summary:**

The authors presented an approach towards extending graphical neural networks to temporal domain using transformers for modeling EHRs. Past research has shown the importance of modeling EHR modalities as GNN that can better capture the inherent correlation better than a flat data structure. However, nuance of EHR data, especially around longitudinal aspects, makes it important to also capture the temporal dynamics. The authors compared their proposed method with several baselines and reported strong results.

**Strengths:**

Some of the key strengths of the paper are as below
- the authors have proposed a hybrid architecture combing GNN and transformer architectures to capture both the spatial and temporal dynamics of EHR data. Key contributions around this proposed architecture are around identifying some of the issues for extending GNN to temporal domain and propose a multi stage pre-training method to capture the dynamics accurately
- The authors compared their methods against strong baselines and reported significant performance improvements
- they key insights around the model being able to capture longer medical histories is very interesting and adds makes the model more applicable
- The authors have also added ablation studies to capture the importance of their training strategy

**Weaknesses:**

The paper can be improved upon by addressing the following aspects
 - The method description can be improved upon. Consider adding an illustration of the training strategy and describing the methods using the illustrations
- the authors can also consider adding sub-group analysis to further strengthen the claims around model performance
- While being cognizant of the page limits, it would have been interesting to analyze some of the inferred graphical networks at individual example level

**Questions:**

See above

---

> ### Author Response · Authors · 2023-11-23
>
> **Improve Method, new illustration**
> We replaced Figure 1 (architecture visualization) with a new improved one. We have also updated the Method section to improve the descriptions. Moreover, we added a new section (Appendix B) containing the new Figure 4, to improve the illustration of the training strategy.
>
> **Add sub-group analysis**
> We have performed a series of new experiments on several patient subgroups (sex and race/ethnicity). We report the results in the Appendix (due to space limitations) and refer to them in the main text.
>
> **Inferred graphical networks at individual level**
> We added a new section (C) in the Appendix to address this suggestion. The section includes Figure 5, which visualizes the learned graphical structure for an individual patient in two different formats (a heatmap and a graph).

---

### Official Review · Reviewer_omW6 · 2023-10-31

**Soundness:** 3 good
**Presentation:** 3 good
**Contribution:** 2 fair
**Rating:** 5
**Confidence:** 3

**Summary:**

This paper focused on patient representations and developed a tailor transformer architecture leveraging both graph transformer and BERT-like encoder only transformer. Evaluation on MIMIC-III and eICU showed improved performance on two tasks, mortality and length of stay.

**Strengths:**

The paper is well written and the developed method is easy to follow.

The architecture sounds reasonable to me.

**Weaknesses:**

Baselines are quite old. The most recent baseline is 2021.More literature research is necessary. For example,  [1,2,3] have dome similar things. Consequently, I am not convinced by the advantage of bringing GNN into BERT,

[1] Hypergraph Transformers for EHR-based Clinical Predictions
[2] EHRSHOT: An EHR Benchmark for Few-Shot Evaluation of Foundation Models
[3] Unsupervised pre-training of graph transformers on patient population graphs

**Questions:**

What do authors would like to convey in the title "BETTER REPRESENTATION IMPROVES DOWNSTREAM PERFORMANCE"
What's the definition of better and how to obtain it?

---

> ### Author Response · Authors · 2023-11-23
>
> **Title clarification**
> We were referring to a “better representation” of EHRs in the embedding space compared to the representations achieved by the current mainstream transformer-based or GNN-based methods (as discussed in the Intro). We added a sentence to the abstract to clarify this. Nevertheless, we appreciate specific suggestions from the reviewer to improve the title.
>
> **Older baselines**
> We have expanded and improved our literature research. We discuss our rationale for the current baselines we picked (pls, see response to 74P1, too). We now added two new (2022 and 2023) baselines (a total of 7), as well as a large new dataset.
>
> Out of the three baselines suggested by the reviewer, we now include the first one (thanks!). The second study (EHRSHOT) came out this summer. Except for a few days, the complete data/model was never released due to the additional approvals needed (pls check the study's GitHub repo). The third study [3] is indeed very relevant, but many methodological details and several key preprocessing steps are missing from the presented code, and despite our efforts, we could not reproduce the results.

---

### Official Review · Reviewer_74P1 · 2023-11-01

**Soundness:** 4 excellent
**Presentation:** 3 good
**Contribution:** 3 good
**Rating:** 6
**Confidence:** 4

**Summary:**

This paper presents GT-BEHRT, an innovative approach that integrates graph-based and transformer-based methodologies to enhance the analysis and predictive accuracy of electronic health records (EHRs). The technique specifically addresses the sparsity in EHR data and the need for capturing complex, graph-type relationships. GT-BEHRT combines graph transformer-derived embeddings for individual visits with a BERT-based framework, facilitating richer patient representations over extended EHR sequences. This method also features a novel two-step pre-training process that further refines the model’s capacity to decode both graphical and temporal patterns in the data. As a result, GT-BEHRT achieves leading performance across diverse medical predictive tasks, indicating its robustness and versatility.

Main Contributions:

GT-BEHRT Design: A new hybrid model that integrates graph transformer and BERT-based architectures to better handle the temporal and graphical nature of EHR data.

Two-Step Pre-training Strategy: A unique pre-training process that enhances the model's ability to understand complex relationships in EHRs, improving performance on predictive tasks.

Superior Predictive Performance: Through its innovative approach, GT-BEHRT sets a new benchmark for state-of-the-art results in various standard medical predictive tasks, demonstrating its potential to significantly impact healthcare analytics.

**Strengths:**

The authors present a commendable effort in intertwining GNN and Transformer methodologies, showcasing an innovative approach to a topic gaining traction in the field. Their literature review is generally thorough, but it's surprising to note the omission of recent contributions from Jure Leskovec's lab, which bear resemblance to this work. While the essence of the paper is intriguing, a deeper exploration of their model's specifics would have provided more comprehensive insights. Overall, the strength of this paper lies in its novel perspective on a burgeoning issue, even if there are areas left to be further elaborated.

**Weaknesses:**

The paper, while advancing an intriguing new architecture, does not fully address the breadth of state-of-the-art works within the Graph Neural Network (GNN) sphere. A more exhaustive acknowledgment and discussion of leading GNN research would have provided a richer context for the authors' contributions. Additionally, while the authors assert that their proposed architecture outperforms existing models, the paper falls short in offering visual depictions of the architecture. Such illustrations are crucial for readers to fully grasp the design and the innovations it purports to bring. Furthermore, the rationale behind the selection of certain models as baselines is not sufficiently elucidated. This lack of detailed justification and visual support may leave the reader questioning the thoroughness of the comparative analysis and the foundation of the authors' claims. The paper would benefit from more comprehensive visual materials and a deeper discussion on model selection to solidify its standing within the current scientific discourse.

**Questions:**

Why did you only used one dataset to test you model?
Have you looked at Michele Moore works?
Why you didn't compare your results with some of the state of the art works such as GMAI?

---

> ### Author Response · Authors · 2023-11-23
>
> **More on GNN research**
> We have expanded our Related Work section to further discuss the leading GNN studies and provide more context on our work.
>
> **Visual depictions of the architecture**
> We replaced the original Fig 1 with a new one, containing improvements to clarify the process. We have also added a new Fig 3 (in the Appendix) to clarify the pretraining process (also part of our contributions).
>
> **Rationale for baselines**
> Out of many available choices, we selected a subset (seven) baselines that i) use either transformer or GNN-based architectures, ii) are recent (2021+) or popular (Dipole), and iii) offer source codes. We also added two new baselines. We added a short note to the paper discussing this.
>
> **Why one dataset**
> We now added the very large and public All of Us dataset. We note that many of the publicly accessible EHR datasets are not quite a good fit, as they contain ICU data (like those on PhysioNet; hence have only short temporal patterns) or do not contain rich code representations for the visits (like Synthea; hence have weak graph representations).
>
> **Moore’s work, SOTA, and GMAI**
> The Perspective study by Moore et al. refers to the growing interest in using foundation models as generalist medical AIs (GMAIs). While very interesting and relevant, only a couple of studies have so far presented an actual model demonstrating generalist-level capabilities (most notably Google’s Med-PaLM). These studies (i.e., those that are truly internet-scale) do not offer their codes (API access also has limitations, especially on patient data). Plus, almost all of these studies focus on unstructured data (like text and images), and none readily uses structured data (like those we use). Inputting EHRs as prompts to such LLMs is doable, but needs separate engineering. We briefly discuss encoder-based (like ours) vs decoder-based (like GMAIs) in the Intro.
> Nevertheless, we now added two new (SOTA) baselines (total 7 baselines).

---

### Author Response · Authors · 2023-11-23

We thank the reviewers for their very careful and elaborate comments. We present separate responses to each reviewer below.

---

### Meta-Review · Area_Chair_yTF4 · 2023-12-06

**Metareview:**

The paper introduces GT-BEHRT, an approach combining graph transformers and BERT models for healthcare predictive tasks using EHRs. It excels in capturing both graphical relationships and temporal sequences in medical data, closely mirroring clinical decision-making. GT-BEHRT achieves top performance in various medical prediction tasks, showcasing its versatility. The proposed method is novel and achieves superior performance based on comprehensive experiments by the authors. Overall, it is a good paper.

**Justification For Why Not Higher Score:**

Reviewers seem not excited enough to make a stronger case.

**Justification For Why Not Lower Score:**

No clear reason to reject this paper.

---

### Decision · Program_Chairs · 2024-01-16

Accept (poster)